# Self-Assembled Permanent Micro-Magnets in a Polymer-Based Microfluidic Device for Magnetic Cell Sorting

**DOI:** 10.3390/cells10071734

**Published:** 2021-07-09

**Authors:** Lucie Descamps, Marie-Charlotte Audry, Jordyn Howard, Samir Mekkaoui, Clément Albin, David Barthelemy, Léa Payen, Jessica Garcia, Emmanuelle Laurenceau, Damien Le Roy, Anne-Laure Deman

**Affiliations:** 1CNRS, INSA Lyon, Ecole Centrale de Lyon, CPE Lyon, INL, UMR5270, University Lyon, Université Claude Bernard Lyon 1, 69622 Villeurbanne, France; lucie.descamps@univ-lyon1.fr (L.D.); marie-charlotte.audry-deschamps@univ-lyon1.fr (M.-C.A.); jordyn.howard@etu.univ-lyon1.fr (J.H.); samir.mekkaoui@univ-lyon1.fr (S.M.); 2CNRS, UMR5306 Institut Lumière Matière, University Lyon, Université Claude Bernard Lyon 1, 69100 Villeurbanne, France; clement.albin@univ-lyon1.fr; 3Laboratoire de Biochimie Et Biologie Moléculaire, Groupe Hospitalier Sud, Hospices Civils de Lyon, 69495 Pierre Bénite, France; david.barthelemy@chu-lyon.fr (D.B.); lea.payen-gay@chu-lyon.fr (L.P.); jessica.garcia@chu-lyon.fr (J.G.); 4CNRS, INSA Lyon, CPE Lyon, CNRS, INL, UMR5270, University Lyon, Ecole Centrale de Lyon, Université Claude Bernard Lyon 1, 69130 Ecully, France; emmanuelle.laurenceau@ec-lyon.fr

**Keywords:** magnetophoresis, micro-magnets, polymer composite, microfluidic devices, particle separation

## Abstract

Magnetophoresis-based microfluidic devices offer simple and reliable manipulation of micro-scale objects and provide a large panel of applications, from selective trapping to high-throughput sorting. However, the fabrication and integration of micro-scale magnets in microsystems involve complex and expensive processes. Here we report on an inexpensive and easy-to-handle fabrication process of micrometer-scale permanent magnets, based on the self-organization of NdFeB particles in a polymer matrix (polydimethylsiloxane, PDMS). A study of the inner structure by X-ray tomography revealed a chain-like organization of the particles leading to an array of hard magnetic microstructures with a mean diameter of 4 µm. The magnetic performance of the self-assembled micro-magnets was first estimated by COMSOL simulations. The micro-magnets were then integrated into a microfluidic device where they act as micro-traps. The magnetic forces exerted by the micro-magnets on superparamagnetic beads were measured by colloidal probe atomic force microscopy (AFM) and in operando in the microfluidic system. Forces as high as several nanonewtons were reached. Adding an external millimeter-sized magnet allowed target magnetization and the interaction range to be increased. Then, the integrated micro-magnets were used to study the magnetophoretic trapping efficiency of magnetic beads, providing efficiencies of 100% at 0.5 mL/h and 75% at 1 mL/h. Finally, the micro-magnets were implemented for cell sorting by performing white blood cell depletion.

## 1. Introduction

Microfluidic magnetophoresis has been demonstrated as an efficient way to trap and separate biological entities and is now integrated in lab-on-chip systems for various biomedical applications, including clinical diagnosis [1,2]. By general principle, biological entities—whether proteins [3,4,5], DNA [6,7,8], or cells [9,10,11]—are magnetically labeled with nano- or microparticles and are dragged towards local magnetic field maxima, generated by magnetic flux sources, with a high selectivity [12].

For a given target magnetic particle, the highest attainable force is limited to the field gradient of the magnetic field source. It is therefore of primary interest to downsize the magnetic field source as it scales up the magnetic field gradients. With micrometer-sized magnets, local magnetic field gradients as high as 10^6^ T/m have been reported in multipole structures [13].

A high density of such micro-traps is needed to efficiently sort entities in fluidic samples. To this end, micropatterning films produced through lithography [14], chemical etching [15,16], or by deposition on topographically structured substrates [17] have been reported. Regardless of the employed microfabrication technique, this film-based approach qualifies as a “top-down” method, offers mastered shapes, and remarkable reproducibility. However, the heterogeneous integration of metals in polymer-based devices raises cohesiveness issues that can be overcome by transferring the synthesized metallic micropatterns in a polymer host matrix [17]. A drastically different approach using a “bottom-up” principle consists of powder compaction and bonding. The host polymer matrix facilitates the integration in polymer-based devices, however, the heterogeneous particle size and morphology lead to less control over the shape and reproducibility of the array of micro-traps [18]. Among the developed approaches, microstructure engineering of powder–polymer mixtures has recently emerged as a noteworthy breakthrough. In principle, the magnetic powder spatial distribution is driven by a magnetic field template during the reticulation of the polymer matrix. It offers a process to reach complementary morphologies, with respect to standard film-based micropatterning processes, without requiring expensive and complex experimental set-up [19].

Here we present a simple fabrication process based on NdFeB particles self-ordering in a polymer matrix, polydimethylsiloxane (PDMS), which is widely used in microfluidic systems. The magnetic forces generated by the NdFeB@PDMS microstructure were simulated in COMSOL and measured with both colloidal probe atomic force microscopy (AFM) and in operando through hydrodynamic determination. We found that magnetic forces up to few nN are reached at contact, and the interaction distance and the magnetization of the target could be increased by adding an external millimeter-sized permanent magnet. This dual-scale permanent magnet was integrated into a microfluidic channel to first measure magnetic bead trapping efficiencies, and then to perform magnetically labeled white blood cell (WBC) trapping.

## 2. Materials and Methods

### 2.1. Materials

A Sylgard Silicone Elastomer (PDMS, Polydimethylsiloxane) was purchased from Samaro (Beynost, France) and consists of two components: a base and a curing agent (10:1 mixing ratio). NdFeB micro-particles are irregularly shaped crushed melt-spun ribbon (MQFP-B, 0.5–7 μm size) provided by Magnequench International, Inc. (Singapore). An SEM image of the NdFeB micro-particles can be found in Appendix A. The magnetic properties of the particles were as follows: remanence, µ_0_M_r_ = 0.9 T (µ_0_ being the permeability of free space); and coercivity field, H_c_ = 740 kA/m. A millimeter-sized permanent magnet (25 × 8 × 2 mm^3^, remanent magnetization µ_0_M_R_~1.4 T, Supermagnete, Gottmadingen, Germany) was also used to study magnetic performance enhancement. Superparamagnetic microbeads (SPMBs) (average diameter: 12 µm, density: 1.1 g/cm^3^, magnetization: 0.66 kA/m, material: magnetite nano-inclusions in a polystyrene matrix, 1 vol% Fe_3_O_4_) were purchased from Kisker Biotech GmbH & Co. (Steinfurt, Germany) to perform colloidal probe AFM measurements and fluidic experiments.

### 2.2. Device Fabrication

#### 2.2.1. Composite Membrane Preparation

Micro-magnet fabrication is based on the composite approach which consists of mixing a hard magnetic powder with a polymer material. NdFeB@PDMS composite membranes were fabricated with concentrations of 2 wt% NdFeB. A flowchart describing the fabrication steps of the micromagnets is reported in Figure 1. Briefly, the mixture containing NdFeB micro-particles and uncured PDMS was poured into a 100-µm-thick Kapton mold stuck to a silanized glass slide [20]. The composite was then cured at 70 °C for 2 h in a magnetic field of 300 mT supplied by a bulk NdFeB magnet (60 × 30 × 15 mm^3^, magnetization along the shortest dimension) to allow PDMS cross-linking and NdFeB particle self-assembly. After Kapton mold removal, the thickness of the composite membrane was increased to 1 mm by pouring liquid PDMS and curing the ensemble at 70 °C for 2 h. Finally, the NdFeB@PDMS membrane was peeled off the glass substrate and NdFeB microstructures were magnetized using a homemade magnetizing system (two magnets of dimensions 50.8 × 50.8 × 25.4 mm^3^, spaced 2.5 mm from each other) that produced a field of 1 T.

#### 2.2.2. Whole Device Assembly

The micro-magnet array was integrated into a microfluidic system by sealing the composite membrane with PDMS microfluidic channels using O_2_ plasma bonding. The channel mold (40 × 0.5 × 0.1 mm^3^) was obtained by soft-lithography using a 50-µm dry photoresist (Eternal Materials Co., Etertec^®^, Kaohsiung City, Taiwan).

### 2.3. Characterization

#### 2.3.1. Structural Characterization

The inner structure of the composite membrane was characterized by X-ray tomography, using the EasyTom Micro and Nano CT Tomography System (RX Solutions). The X-ray source was a LaB6 cathode with a diamond window leading to higher flux (20 μA). Its focal spot measured 0.25 μm and a tension of 90 kV was applied. Scans were acquired by a CCD detector, whose matrix measured 2000 × 1312 pixels, with a voxel resolution of 0.3 μm^3^. 3D images were reconstructed from projections at 1400 different angular positions. Final images of 1700 × 1700 × 400 voxel, i.e., 510 × 510 × 120 μm^3^, were obtained and processed with ImageJ to characterize NdFeB particles’ spatial organization in the volume of the composite membrane. Conventional optical microscopy characterizations using an Olympus BX51M microscope coupled to a camera (Moticam2000, Motic^®^, Wetzlar, Germany) were also carried out with ImageJ to study the in-plane organization of the NdFeB agglomerates.

#### 2.3.2. Magnetic Characterization

Magnetization and demagnetization processes were measured in a SQUID magnetometer (Quantum Design MPMS XL). The characterized sample was a 100-µm-thick, 4-mm-sided square piece of NdFeB@PDMS membrane, bonded to a 1-mm-thick PDMS support. The columnar NdFeB agglomerates pointed along the membrane’s thickness. The sample was fixed in a straw perpendicularly using Kapton tape. Therefore, the applied magnetic field in the SQUID was directed along the long axis of the NdFeB chain agglomerates, similar to the set-up of the operational device.

#### 2.3.3. Magnetic Force Measurements

Magnetic force measurements were carried out on the AFM MFP-3D (Asylum Research, Oxford Instrument, Goleta, CA, USA) using a 15-µm superparamagnetic colloidal probe (SPMB) glued to a silicon nitride cantilever (PNP-TR-TL, stiffness measured with thermal noise method: 43 pN/nm, NanoAndMore, Paris, France). The acting force between the colloidal probe and the sample was recorded as the probe approached and withdrew from the sample surface. In particular, the sample was scanned in two-pass mode, as follows: the first scan was performed at contact, to localize the micro-magnets, and the second one at a distance of a few hundred nanometers, to record the cantilever’s deflection above the sample surface. In this way, the maximum force generated by the localized micro-magnet can be measured, and the approach/retract curve is performed at the exact micro-magnet position (see Appendix A) at a 1 µm/s constant velocity. Measurements were performed in a 10 mM NaCl solution to screen electro-static forces. A total of 40 micro-magnets were characterized using this method.

### 2.4. Bead Injection

SPMBs were suspended in a filtered phosphate buffer saline (PBS, Sigma-Aldrich, Saint-Louis, MO, USA) solution with 2% pluronic F-108 (Sigma-Aldrich, Saint-Louis, MO, USA) at a concentration of 50 SPMBs/µL and injected into the microchannel using a pressure-driven flow controller (Flow-EZ™, Fluigent, Le Kremlin-Bicêtre, France). A picture of the experimental set-up is provided in Appendix A.

### 2.5. Cell Sample Preparation

WBCs were collected from a whole blood sample after red blood cell lysis and were magnetically labeled with superparamagnetic nanoparticles (Ademtech SA, Pessac, France, MasterBeads Carboxylic Acid 0215). The nanoparticles were 500 nm in diameter and composed of a magnetic core (approximately 70% iron oxide) encapsulated by a hydrophilic polymer shell with carboxyl groups on its surface. They were functionalized with both anti-CD45 and anti-CD15 antibodies (purchased from R & D Systems, Minneapolis, MO, USA) to enhance WBC magnetic labeling. Finally, WBCs were suspended in 500 µL of EDTA (2 mM, PBS-BSA 2%) at a concentration of 6.10^5^ WBCs/mL. PC-3 cancer cells were spiked into the white blood cell sample, at a concentration of 4.10^4^ PC-3/mL, to study the specificity of the trapping. WBCs were stained with Hoechst 33342 (Ready Flow Reagent™, ThermoFisher Scientific, Waltham, MA, USA) and PC-3 cells were tracked with a green dye (CellTracker™ Green CMFDA Dye, ThermoFisher Scientific, Waltham, MA, USA).

## 3. Results and Discussion

### 3.1. Micro-Magnet Microstructure and Magnetic Performances

The reconstructed 3D profile of the NdFeB@PDMS membrane from X-ray tomography observation is shown in Figure 2a, revealing a predominantly chain-like organization of NdFeB micro-particles in the PDMS matrix. This chain organization is particularly visible in Figure 2b, where a 100-µm-thick slice of the composite is presented in the (y, z) plane. During the PDMS cross-linking step, in which an external magnetic field is applied, the NdFeB particles, seen as magnetic dipoles, are driven by dipole–dipole interactions [21]. A nearly homogeneous applied field favors a uniaxial stacking of the particles along the field direction and a repulsion in the perpendicular directions [22]. The X-ray tomography images of the self-assembled microstructures were characterized using ImageJ. The average length of the microstructures is in the order of 75 µm and the interparticle distance can reach 0.4 to 4 µm for some chains, knowing that the resolution of X-ray tomography is 0.3 µm. Although the chains present a certain heterogeneity, the composite approach allows the integration of micro-magnet arrays in PDMS without complex and costly technological steps. A top view of the micro-magnets is presented in Figure 2c using brightfield microscopy. Their diameter ranges between 0.6 and 18 µm, of which 98% are smaller than 10 µm. The nearest neighbor distance (nnd) distribution is shown in Figure 2d, with an average distance of 16 µm. In general, the Poisson law can describe independent events and has no adjustable parameters. Applied to a given number of particles on a known surface, it allows the description of a random distribution. Here, it fails to describe the experimental nnd distribution as obtained from brightfield microscopy observation. It is better described by a normal distribution, demonstrating that the self-organization of the micro-magnets is not random. The total density of the magnetic microstructures reached 1250 ± 130/mm^2^.

Prior to its implementation in a microfluidic device, the magnetization of the NdFeB@PDMS composite membrane was measured in a SQUID magnetometer at room temperature. The out-of-plane magnetization loop is shown in Figure 3a, specifically when the field is applied along the agglomerates’ long axis. In order to determine the operating magnetization state, first magnetization curves were measured with successive returns to the remanent state. The first point, M_RI_ (initial remanent magnetization), indicates the remanent magnetization after the field reticulation process. M_RF_, corresponding to the full remanent magnetization, was 0.75 M_S_ (µ_0_M_RF_ = 0.84 T), and was reached with a field larger than 2.5 T. M_RO_ (operating remanent magnetization) corresponds to the magnetization after magnetizing the membrane in a 1 T field, which is equivalent to the field used in the magnetizing process of the operating device. M_RO_ reached 0.48 M_S_, and we can estimate the remanent magnetic field of the micro-traps, µ_0_M_RO_, to be 0.54 T.

The magnetic properties of the organized NdFeB micro-magnets in the PDMS matrix were modeled using a finite element approach (COMSOL, AC/DC module, 2D axisymmetric model). Based on X-ray tomography observations, an individual chain-like structure was modeled as 28 cylindrical NdFeB particles of diameter 4 µm and height 1.5 µm, each spaced apart by 1.2 µm (total chain length of 75 µm). From SQUID measurements, the remanent field of the particles was set at 0.54 T. The magnetic field and magnetic field gradient were calculated as a function of the distance from the composite surface (Figure 3b). The magnetic field gradient generated by the magnetic structure, along the z-direction, is pictured by a magnetic coefficient (Cm,z), defined as follows:(1)$$\mathrm{Cm},z=\frac{1}{\sqrt{B_{r}{}^{2}{\;+\;B}_{z}{}^{2}}}\left(B_{r}\frac{\partial B_{z}}{\partial r}{\;+\;B}_{z}\frac{\partial B_{z}}{\partial z}\right)$$

The expression of Cm,z was obtained from the development of the expression of the magnetic force (see the Appendix A).

The microstructure generates a magnetic field of 200 mT and a magnetic field gradient of 10^5^ T/m at its surface, which are adequate with regard to numerical values found in the literature [23,24,25]. In turn, the magnetic field and the magnetic field gradient then decreased with the distance to the microstructure, down to 0.1 mT and 5 T/m, respectively, at a distance of 50 µm (Figure 3b). Magnetic objects initially flowing at the top of the channel may not be subjected to the relatively short interaction range of the micro-magnets, limiting the trapping efficiency in a 100-µm-high channel. Therefore, we quantified the effect of adding an external millimeter-sized permanent magnet under the composite membrane (at a distance of 1 mm from the micro-magnets). Thus, as compared with the configuration with micro-magnets only, the combined use of the micro-magnets and milli-magnets doubled the magnetic field value in contact and increased the minimum value inside the channel from 0.02 mT to 200 mT. The higher field value ensures a higher magnetic moment held by the target objects. Furthermore, the magnetic field gradient increased from 5 T/m to 40 T/m at a distance of 50 µm (Figure 3b), therefore increasing the interaction distance [12].

### 3.2. Generated Magnetic Forces

We used colloidal probe AFM to measure the magnetic force produced by individual micro-traps. The SPMB, attached to the tipless AFM cantilever, was approached, then retracted from the composite’s surface. The attractive force acting on the probe, measured by the cantilever deflection, is linked to the magnetic force. Contact forces ranging from 0.5 to 2 nN were measured (Figure 4a), and these values are in agreement with the data found in the literature for microscale soft magnetic sources [26,27,28] and hard magnetic structures [29]. The mapping of the magnetic attraction, performed at a distance of 500 nm from the surface, highlights that the maximum force is localized above the micro-magnet (Figure 4a, inset). Regarding the influence of an external permanent magnet, the magnetic force range was increased to 4 nN. Indeed, the magnetic force acting on the superparamagnetic bead of magnetization $M_{b}$ and volume $V_{b}$ is given by $\vec{F_{\mathrm{mag}}}=\mu_{0}V_{b}\left(\vec{M_{b}}\cdot \vec{\nabla }\right)\vec{H}$, where H is the applied magnetic field. The magnetization of SPMB (see Appendix A), when positioned at the composite’s surface (i.e., in a field of 200 mT), is not fully saturated, and so the higher magnetic field value with the addition of the milli-magnet drove it up by 10% (from 470 to 510 A/m, Appendix A). Similarly, the magnetization of the micro-traps was slightly raised by 5% (from 490 kA/m to 515 kA/m, Appendix A).

Considering the low SPMB concentration, we used the single-particle-transport model to analyze the involved forces [30]. Moreover, given the low magnetic moment of the micro-beads, we neglected the dipole interactions and the contribution of their residual magnetization. Due to the micrometric size of the SPMBs, mass diffusion and magnetic diffusion based on Brownian motion can be ignored [31]. Finally, the buoyancy and gravity exerted on SPMBs are negligible, as compared with the magnetic and drag forces [2]. Thus, in the magnetic microfluidic system, the two main forces of importance to evaluate trapping efficiency are the magnetic force and Stokes’ drag force (Figure 4b). Reliable trapping is obtained when the magnetic force value exceeds that of the drag force. We estimated the magnetic force generated by the micro-magnets, in operando, through the hydrodynamic determination of the holding magnetic force [19,32]. SPMBs were injected at an equivalent flow rate of 2 mL/h and trapped on micro-magnets. Then, we injected PBS and gradually increased the flow rate and identified, for each trapped bead, the flow velocity, and thus the shear stress, at which it was untrapped. Average forces of 1.3 ± 0.5 nN were measured on more than a thousand beads, which is in good agreement with the forces measured by colloidal probe AFM. We also observed that traps located at the beginning of the trapping area generated greater forces than those located in the middle of the trapping zone. In particular, those located in the first 50 µm of the trapping zone generated forces on average 1.8 times greater than those produced overall in the trapping zone, as shown in Figure 5a. This observation is in good agreement with the finite element simulation calculations of the field gradients generated by a network of chains (COMSOL, AC/DC module, 2D model). We first determined the number of chains from which the network can be considered as an infinite number of chains, ensured by a variation of Cm intensity lower than 1%. This threshold was found to be nine chains (see Appendix A). Then, the magnetic field gradient generated above a network of nine chains was compared with the one produced by a single chain (Figure 5b). 2D calculations show that a chain included in a network produces a Cm that is 34% lower than an isolated chain. As for the broken symmetry at the edge of the network, its effect on Cm is significant over a distance of about 50 µm from the edge and the Cm produced by a chain on the side is 15% greater than that of a chain included in the network. These observations show the demagnetizing interaction between neighboring chains which tends to decrease the generated magnetic force. This demagnetizing effect inevitably occurs in any dense micro-array of magnetic structures. In these types of applications, there is always a tradeoff between the density of the micromagnets and their individual trapping efficacy.

We also compared the forces generated by the same trap in the presence or absence of the external magnet. Although these measurements were performed with different trapped beads, whose diameters varied from 8 to 20 µm (see Appendix A for SEM image of the SPMBs), we observed, on 192 traps, that for 70% of them the magnetic forces were greater or equal in the presence of the magnet.

### 3.3. Implementation of the Micro-Traps in Microsystems for the Manipulation of Magnetic Entities

In order to evaluate the sorting potential of micro-magnets for biomedical applications, we first evaluated the trapping efficiency of SPMB models and then demonstrated the trapping of magnetically functionalized cells. The magnetophoretic trapping efficiency of the micro-magnets integrated into the microfluidic system was determined on a sample of 6500 SPMBs at various flow rates (Figure 6a). At 0.5 mL/h, 85% and 100% of injected beads are trapped in the absence and in the presence of the milli-magnet, respectively. At this flow rate, the trapping throughput is about 400 beads/min. The trapping area, being 7.5 mm^2^, has a trapping density of 750 beads/mm^2^ when all beads are trapped. At 1 mL/h, the improvement from adding milli-magnets is even more pronounced. The trapping efficiency reaches 75% with the combination of micro- and milli-magnets, whereas it is 50% with the micro-traps alone. Indeed, from COMSOL calculations of the magnetic field value along the 100-µm channel height (Figure 3b), the magnetization of the flowing SPMBs is significantly increased throughout the channel in the presence of the milli-magnet. Besides, as shown previously by COMSOL simulations, at a distance greater than 50 µm above micro-traps, the magnetic field gradient generated by the milli-magnet predominates (Figure 3b) and allows flowing targets located in the upper part of the channel to be dragged down. The strong and localized magnetic field gradients generated by the micro-magnets then efficiently retain the trapped targets. It is worth mentioning that the milli-magnet alone leads to significantly less trapping efficiency, as compared with the micro-magnets. Indeed, the generated gradients at the channel surface are more than three orders of magnitude lower than those generated by the micro-magnets (30 T/m vs. 10^5^ T/m), which results in a trapping efficiency of only 48% at 0.5 mL/h with the milli-magnet alone. It is therefore the combination of the two types of magnet that allows higher magnetic forces to be generated, which explains the higher trapping efficiencies. Other works rely on the association of passive and magnetic functions to attract the targets towards the magnets. For example, Chung et al. implemented herringbone structures on the top of the channel to deflect magnetic objects to the surface of the magnetic traps by chaotic mixing [33]. Beyond 2.5 mL/h, the benefit of the external milli-magnet becomes negligible in comparison with the drag force. Thus, circulating micro-beads which are far from the traps are no longer attracted to the latter with the external magnet. The trapping performances are therefore identical for both configurations.

Finally, the biocompatibility and operation of the trapping device were established with biological samples by successful WBC depletion. Specific capture of magnetically labeled WBCs on the micro-traps is illustrated in Figure 6b. Spiked PC-3 cancer cells were recovered at the output; therefore, the device could be easily implemented for cell sorting based on their magnetic functionalities.

## 4. Conclusions

We demonstrated an original approach to integrate large density arrays of permanent micro-magnets into a microfluidic system. This approach, based on hard magnetic powder–polymer composites, led to autonomous and compact systems that were successfully implemented for the manipulation of SPMBs and the sorting of magnetically labeled cells. The integrated micro-magnets can generate magnetic forces of several nN, which is comparable to the magnetic force generated by micro-scale magnetic sources obtained by more conventional and complex microfabrication methods. We identified high magnetic field gradients generated by the integrated micro-magnets of up to 10^5^ T/m at their surface. We used an external milli-magnet to add a background field throughout the channel and quantified its effect on magnetic forces and trapping efficiency. This was conducted using three approaches: numerical simulations, colloidal probe AFM measurements, and fluidic experiments. In particular, we demonstrated that this additional field led to an increase in the trapping efficiency for flow rates of between 0.5 and 2.5 mL/h. At 0.5 mL/h, the trapping efficiency rose from 85% to 100%. Moreover, cell isolation was performed through the depletion of magnetically labelled white blood cells. These cost-effective functional materials integrating micro-magnet arrays open the way to a broad range of magnetophoretic applications, especially in the biomedical field: from the detection of biological molecules via the manipulation of functionalized magnetic microbeads, to immunomagnetic separation requiring the manipulation of thousands of target objects in a blood sample.

## Figures and Tables

**Figure 1 cells-10-01734-f001:**
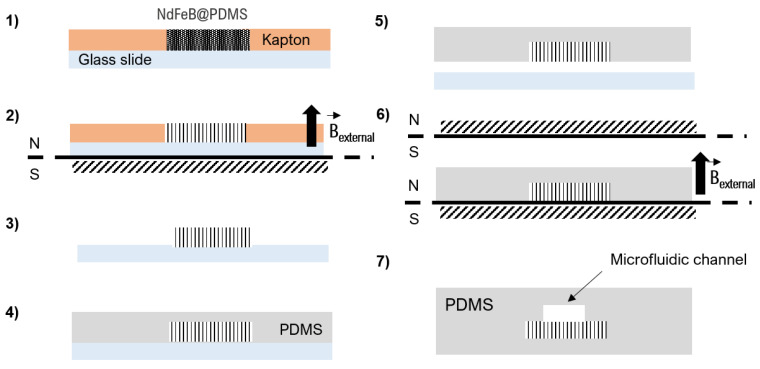
Fabrication steps of the microdevice: (**1**) the composite is molded in a Kapton film bonded to a silanized glass slide substrate; (**2**) the composite is then placed in a 300 mT magnetic field for NdFeB particle self-organization in chains at 60 °C for 2 h; (**3**,**4**) the Kapton mold is then removed and pure PDMS is poured; (**5**) after curing at 70 °C for 2 h, the composite membrane is peeled off and (**6**) magnetized under a magnetic field of 1 T. Finally, (**7**) the composite membrane is bonded to a channel molded in PDMS by O_2_ surface plasma activation.

**Figure 2 cells-10-01734-f002:**
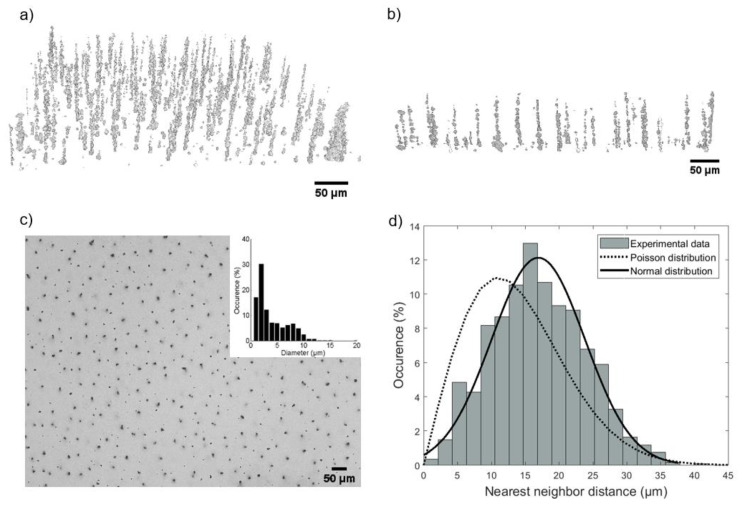
(**a**) 2% NdFeB particle organization in PDMS membrane from X-ray tomography reconstruction of a 510 × 510 × 120 μm^3^ volume. (**b**) Section view of the chain-like organization of the 2% NdFeB particles obtained from X-ray tomography reconstruction of a 100 × 510 × 120 µm^3^ volume. (**c**) Microscope image of the micro-magnet array (top view), and (inset) the micro-magnet diameter distribution. (**d**) Nearest neighbor distance distribution. The experimental distance, measured from brightfield microscopy observation (0.8 mm^2^ images), followed a normal distribution and not a Poisson distribution (random distribution of N = 1225 particles in a 0.8 mm^2^ frame).

**Figure 3 cells-10-01734-f003:**
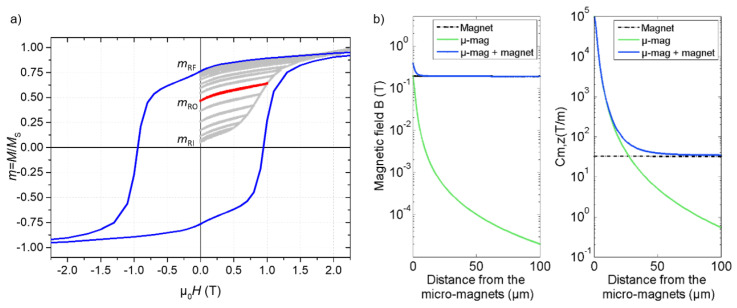
(**a**) Magnetization of the NdFeB@PDMS membrane, measured out-of-plane. The light grey curve is the first magnetization curve with successive measurements of remanent magnetization. The thick red curve highlights the remanent magnetization at 1 T. The blue curve is the full magnetization loop. (**b**) COMSOL simulations of the magnetic field and magnetic field gradient (defined by Cm,z) generated by the micro-magnets (µ-mag) as a function of the distance from the composite surface, with or without an external magnet below the composite surface.

**Figure 4 cells-10-01734-f004:**
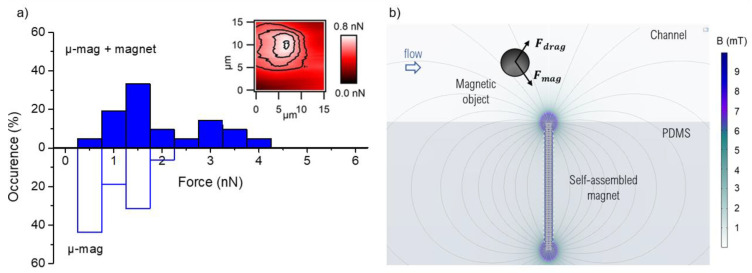
(**a**) Distribution of the magnetic force produced by the micro-magnets (µ-mag) from colloidal probe AFM measurements, with or without the external milli-magnet. The inset shows an example of a second scan in two-pass mode, at 500 nm above a micro-magnet (no external magnet), in order to localize the force of maximum intensity. (**b**) Schematic representation of the forces acting on a magnetic target flowing in a microfluidic device with magnetic structures located at the bottom of the channel.

**Figure 5 cells-10-01734-f005:**
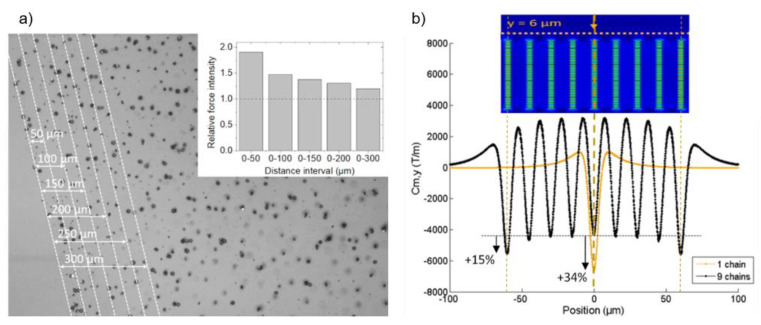
(**a**) Calculation of the magnetic coefficient generated above a network of nine chains and comparison with a single chain. (**b**) Relative force intensity according to the position in the microchannel. The relative force intensity refers to the ratio of the measured force in one position in the channel to the average force in the whole channel.

**Figure 6 cells-10-01734-f006:**
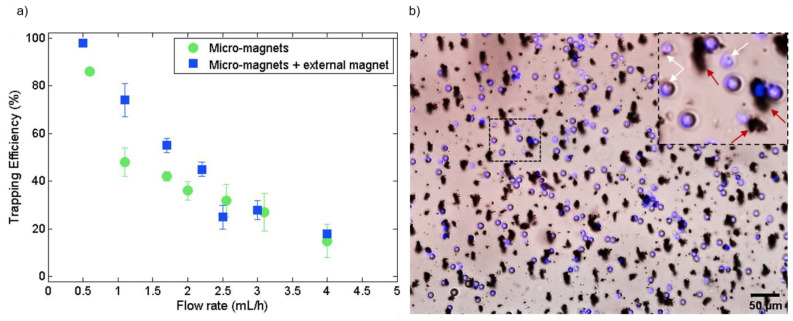
(**a**) Magnetic trapping efficiency of 12-µm SPMBs on the micro-magnets as a function of the flow rate, with or without the external magnet. (**b**) Composite microscope image showing trapped magnetically labeled WBCs (blue) in the microfluidic channel at 200 µL/h. All white blood cells are trapped on magnetic traps. What appears to be large traps are instead agglomerates of magnetic nanoparticles (red arrows in the inset). The inset highlights the trapping of white blood cells on less visible magnetic traps (white arrows).

## Data Availability

Not applicable.

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
