# Peer review of "Self-Assembled Permanent Micro-Magnets in a Polymer-Based Microfluidic Device for Magnetic Cell Sorting"

_cells, 2021, doi:10.3390/cells10071734_

Round 1

Reviewer 1 Report

Introduction

The article uses experiment, numerical simulation (COMSOL) and colloidal probe AFM (Atomic Force Microscope) to study magnetophoresis for the separation of biological entities in polymer-based microfluidics. The efficacy of their micro-magnetic systems was studied based on how it affects the trapping efficiency of the target material at different flow rates.

Primary Concern: The computational analysis of magnetophoresis does not include all the possible forces acting on SPMB. For example, see the article - Ayansiji, Ayankola O., et al. "Constitutive relationship and governing physical properties for magnetophoresis." Proceedings of the National Academy of Sciences 117.48 (2020): 30208-30214.

Authors have only considered the magnetic convection and drag forces, but not the magnetic diffusion, residual magnetization force, and possible migration forces due to charges on SPMB. I recommend authors calculate the values of these forces for their system and justify why magnetic convection and drag forces are dominant (if this is the case).

Suggestions

  • In line 24, the full form of AFM, (Atomic Force Microscope) should be presented the first time it is used in the paper.
  • In Line 99, what is the homemade magnetizing system?
  • Provide an explanation to the irregularity observed for the trapping efficiency when both micro magnet and external magnet are used at the flow rate of 2.5 mL/h (Figure 5a)
  • Provide insight into why the additional external magnetic force has no effect on the trapping efficiency at the flow rate of 2.5mL/h and beyond
  • Can this model be extended to the case of micromagnets with external fields?

Reviewer 2 Report

To attract magnetic microbeads on a microchannel wall, the authors produced high magnetic field gradients by self-assembled micro columnar magnets. Cell capture was demonstrated and it was interesting. This reviewer recommend the publication of this manuscript, but there are a few points that concerned to this reviewer.

  1. In figure 5(a), no results with only external magnet is shown. Only in the text, 48% efficiency is mentioned. The results with only external magnet should be included in figure 5(a).
  2. About figure 5(b), cells are expected to be captured on the edge of micro magnets. It seems that there are many cells that were captured between magnets. Quantitative discussion may be helpful for audience.
  3.  Figure S2. should be included in the main parts, and "Bext" is misleading. More appropriate description, such as Bexternal, would be preferable.
